# ECG Forecasting System Based on Long Short-Term Memory

**DOI:** 10.3390/bioengineering11010089

**Published:** 2024-01-17

**Authors:** Henriques Zacarias, João Alexandre Lôbo Marques, Virginie Felizardo, Mehran Pourvahab, Nuno M. Garcia

**Affiliations:** 1Faculdade de Ciências de Saúde, Universidade da Beira Interior, 6201-001 Covilha, Portugal; 2Instituto de Telecomunicacoes, 6201-001 Lisboa, Portugal; virginie@it.ubi.pt (V.F.); nmgarcia@ciencias.ulisboa.pt (N.M.G.); 3Instituto Politécnico da Huíla, Universidade Mandume Ya Ndemufayo, Lubango 1049-001, Angola; 4Laboratory of Applied Neurosciences, University of Saint Joseph, Macau 999078, China; alexandre.lobo@usj.edu.mo; 5Departamento de Informática, Universidade da Beira Interior, 6201-001 Covilha, Portugal; m.p@ieee.org; 6Faculdade de Ciências, Universidade de Lisboa, 1749-016 Lisboa, Portugal

**Keywords:** electrocardiogram, long short-term memory, forecasting

## Abstract

Worldwide, cardiovascular diseases are some of the primary causes of death; yet the early detection and diagnosis of such diseases have the potential to save many lives. Technological means of detection are becoming increasingly essential and numerous techniques have been created for this purpose, such as forecasting. Of these techniques, the time series forecasting technique seeks to predict future events. The long-term time series forecasting of physiological data could assist medical professionals in predicting and treating patients based on very early diagnosis. This article presents a model that utilizes a deep learning technique to predict long-term ECG signals. The forecasting model can learn signals’ nonlinearity, nonstationarity, and complexity based on a long short-term memory architecture. However, this is not a trivial task as the correct forecasting of a signal that closely resembles the original complex signal’s structure and behavior while minimizing any differences in amplitude continues to pose challenges. To achieve this goal, we used a dataset available on the Physio net database, called MIT-BIH, with 48 ECG recordings of 30 min each. The developed model starts with pre-processing to reduce interference in the original signals, then applies a deep learning algorithm, based on a long short-term memory (LTSM) neural network with two hidden layers. Next, we applied the root mean square error (RMSE) and mean absolute error (MAE) metrics to evaluate the performance of the model and obtained an average RMSE of 0.0070±0.0028 and an average MAE of 0.0522±0.0098 across all simulations. The results indicate that the proposed LSTM model is a promising technique for ECG forecasting, considering the trends of the changes in the original data series, most notably in R-peak amplitude. Given the model’s accuracy and the features of the physiological signals, the system could be used to improve existing predictive healthcare systems for cardiovascular monitoring.

## 1. Introduction

The human body produces a set of biological processes from which we can measure and collect signals that can be represented as time series. Different processes produce different signals, which distinguish a large range of physiological events, whose characteristics differ from case to case. As examples of such signals, the heartbeat can be calculated by, e.g., analyzing an electrocardiogram (ECG), and brain waves are usually represented by an electroencephalogram (EEG); yet there are many types of physiological processes that can be analyzed using other types of measurements, such as those represented by images or the concentration of chemicals in tissues or fluids.

The pervasiveness of computational devices, many of which can be wearable, has resulted in an increase in the use of devices that integrate sensors, including Internet of Things (IoT) devices, which are often used to collect such signals. Many of these devices allow for the registration and storage of large amounts of data and their re-utilization for several different lines of research [1].

Despite the wide variety of these devices, most are available at affordable prices. According to the World Health Organization (WHO) [2], it is estimated that 17.9 million people died from cardiovascular disease (CVD) in 2019, which represents 32% of all deaths worldwide and makes it the leading cause of death across the globe [2,3]. It is reasonable to assume that at least some of these deaths could have been prevented by timely diagnosis using ECG trace analysis, which is considered to be the gold standard of cardiovascular monitoring and offers a wealth of diverse information.

The interpretation of ECGs can be a complex task involving visual inspection; even so, the analysis of ECGs is routinely performed by specialized health professionals to assess cardiac health status. Recently, the use of computer-aided diagnosis systems has become regarded as a way to carry out the task of detecting or classifying CVD [4,5,6,7], at least in the initial screening stage.

Therefore, the early detection of anomalies in ECG signals, possibly using computer-aided diagnosis as a prediction mechanism for the occurrence of CVD, is a powerful tool for healthcare systems worldwide as it may help healthcare professionals in their decision-making processes or possibly to prevent dangerous clinical events [4,5,8]. Different approaches can be followed in order to predict or detect issues regarding ECG signals [3,9,10].

One of the techniques used for this is the forecasting approach. The time series forecasting methodology is used to make predictions about forthcoming events [11]. The use of time series forecasting methods to predict physiological data has the potential to help healthcare providers to anticipate and mitigate the need to treat patients depending on their diagnosis [12,13].

Over recent years, the number of cardiovascular disease forecasting studies has grown and is now substantial [14]. However, to our knowledge, relatively few works have been explicitly related to ECG signal forecasting. This constraint presents many benefits, including originality and possible noteworthy advancements in the domain of cardiovascular disease (CVD) detection and prediction, particularly in the exploration of uncharted challenges. However, it also entails drawbacks, such as the absence of established methodologies, limited benchmarking opportunities, and restricted guidance.

As previously indicated, electrocardiogram (ECG) analysis is the conventional instrument for monitoring and assessing cardiac activity. The ability to forecast electrocardiogram (ECG) patterns plays a crucial role in the identification and diagnosis of various cardiac ailments, including arrhythmias, myocardial infarction, heart blockages, and other significant heart-related problems. Additionally, this ability has the potential to make substantial contributions to the field of personalized medicine and preventative healthcare, mitigate the likelihood of cardiovascular problems, and enhance the overall quality of life of patients.

The main objective of this paper is to present a model that uses a deep learning technique based on a long short-term memory (LSTM) architecture to forecast long-term ECG signals, which is able to learn the nonlinearity, nonstationarity, and complexity of ECG signals. It can also possibly be used to create effective forecast signals with identical structures and behaviors to the original signals, with small differences in amplitude.

The rest of this paper is organized as follows: Section 2 presents a literature review that embodies the work in this area; Section 3 presents the materials and methods used to address ECG forecasting; Section 4 presents the implementation of the proposed method; Section 5 presents a discussion of the results; Section 6 concludes the paper.

## 2. Literature Review

The scientific community has long been faced with the challenges of exploring machine learning algorithms for the processing of ECG data. Over the past decade, researchers have extensively utilized sophisticated machine learning algorithms to extract significant insights from ECG data. The MIT-BIH database has been a fundamental resource for these studies, offering a reliable and extensively used framework for assessing the effectiveness of different machine learning models. The MIT-BIH database has significantly influenced the field of machine learning-based ECG analysis for over 10 years. Scientists have utilized this large dataset to create and improve algorithms that can increase the accuracy and efficiency of automated ECG interpretation [15,16].

Forecasting techniques predict future events by analyzing past trends, on the assumption that future trends will remain similar to those in the past [17,18]. The basic idea is to extend underlying trends using predictable trials that are already in the data.

To deal with real-time series in modern applications, several methods are used, including statistical approaches [17,18,19,20,21], machine learning techniques [22,23,24,25,26], and deep learning architectures [6,27,28,29,30]. These methods have been investigated and applied by the academic community in several areas, such as the economic and financial [24,26,31,32,33,34], industrial [22,31,32,35], scientific [29,33], and healthcare [7,14,18,34,36,37,38] fields. Nevertheless, within healthcare, and considering the use of continuous physiological data in particular, there has been a very limited number of works on forecasting [6,7], with the possible exceptions of forecasting diabetes and COVID-19 [12,13]. Although there are limitations in the current body of research on ECG forecasting, the available literature does include some studies that have used hybrid methodologies, such as artificial neural networks and deep learning models.

In their study, Mohammadi, Ghofrani, and Nikseresht [39] integrated a forecasting framework that incorporated ridge regression, high-order fuzzy cognitive maps (HFCMs), and empirical wavelet transform (EWT). They conducted experiments using 15 distinct time series datasets, one of which comprised 4170 samples of ECG data. However, their paper lacked some details; in particular, the specific lock-back length used in the analysis. The authors reported that the experiment successfully attained a root mean square error (RMSE) of 0.011.

Sun et al. [40] proposed a method for predicting new ECG data based on a variational mode decomposition (VMD) and back-propagation (BP) neural network, using one ECG record retrieved from the MIT-BIH arrhythmia database. The authors extracted 15 segments from the signal, each with 2160 samples, and then split them into training, validation, and test (60-20-20) sets, reportedly achieving an RMSE of 0.00233 and a mean absolute error (MAE) of 0.0157. The same researchers conducted another study with the same goal using a similar methodology, changing only the VMD to mutual information, resulting in RMSE and MAE values of 0.0423 and 0.024, respectively [41].

A novel ECG signal prediction method based on the autoregressive integrated moving average (ARIMA) model and discrete wavelet transform (DWT) was proposed by Huang et al. [42]. The authors used 2768 samples from four different ECG records in the MIT-BIH arrhythmia database. These samples were then split into training and test sets (900 and 1868 samples, respectively). They obtained an RMSE of 0.0180775 and an MAE of 0.011105.

In another study by Prakarsha and Sharma [43], an artificial neural network was applied to forecast ECG time series. To validate the model, two ECG signals were used: one was simulated using the Neurokit tool and the other was a real ECG signal retrieved from the Physio net ATM of sleep apnea. The number of samples used was 20,000, which were split 75/25% for training and testing. A slide windows technique with a sequence length of 64 samples was used and authors reported an MAE of 0.045.

Forecasting studies based on deep learning techniques have become increasingly present in the literature [6,32,44] and here, we highlight cases related to ECG signals. The work of Dudukcu et al. [45] proposed a temporal convolutional network (TCN) approach with an RNN for chaotic time series. In this study, TCN models were tested. In the first model, LTSM was used in the initial layers of the model and a TCN was applied in the fully connected layers, while in the second model, a TCN was considered in the initial phase of the GR network. The research included nine different time series, including an ECG series, for which 21 records were selected from the MIT-BIH arrhythmia database. In total, 100,000 samples were divided into training, validation, and test sets for each record (40,000-10,000-50,000), considering a sequence length of 10. The RMSE and MAE metrics were calculated and the following results were obtained: 0.0082 and 0.0051 for the first model; 0.0084 and 0.0052 for the second model.

Further, Festag and Spreckelsen [46] presented a study on the imputation and forecasting of multivariate medical time series based on a recurrent conditional Wasserstein GAN and attention. Data from the Autonomic Aging Physio net (blood pressure series and ECG) were used. For the study, 411 records were selected (training = 275; testing = 6; validation = 65) and each record was segmented into up to 10-s samples. In the end, 26,939 samples were used in training, 5972 were used in validation, and 6062 were used in testing. Results were obtained separately for each series and three different forecasting scenarios were considered: (a) an output in one step with an error (MSE) of 0.0429; (b) an output of 30 steps with an MSE of 0.0858; (c) a forecast horizon of 250, with an MSE of 0.0891. Table 1 serves as a thorough summary of the main cited articles, offering a consolidated overview of the employed techniques, the numbers of utilized ECG signals, the lengths of the sequences, and the values of the metrics used.

Although the use of deep learning (DL) algorithms is promising in forecasting tasks, the number of studies that have applied them to ECG signals remains small, which opens up the opportunity to explore different algorithms.

## 3. Materials and Methods

### 3.1. ECG Data Description

The MIT-BIH database was selected to conduct this study. This database is public and available from Physiobank [47]. It has been presented in most publications found in the literature involving ECG signals [48,49]. The database contains 47 heartbeat records from different patients, each covering 30 min.

Each record contains two ECG leads. In most samples, the principal lead (lead A) is a modification of lead II (electrodes on the chest). The other lead (lead B) is usually a modified lead V1, but in some records, this lead is V2, V5, or V4 [47,50]. These ECG signals were sampled at 360 samples per second, with an 11-bit resolution, over a 10-mV range, and bandpass-filtered at 0.1–100 Hz.

Independent cardiologists annotated and verified the records’ timings and beat class information. The beat classes include normal beat (N), left bundle branch block (LBBB), right bundle branch block (RBBB), atrial premature beat (APB), and premature ventricular complex (PVC), which are summarized in Table 2. Only lead A was selected for consideration in this study.

### 3.2. Proposed Methods

#### Pre-Processing

Since most ECG readings also include background noise and other interference, a means of obtaining a cleaner representation of signals is desirable.

Pre-processing is a method used to improve final signals by bringing them closer to their raw forms. Pre-processing ECG readings is performed so as to remove as many potential artifacts as possible, such as noise, interference, and other phenomena. The ECG signals were smoothed using the simple moving average (SMA) method, with a window size of 3, to remove noise from patients’ movements or ECG sensors’ muscle activity. Then, a median filter with a large initial step was used to reduce the signals’ obnoxious noises via a nonlinear digital filtering method, in which the median of neighboring values was used to replace the original value. Next, the ECG signals were sent through a notch filter to filter out any remaining power line noise (60 Hz). The pre-processing phase ended by normalizing the signals to the range from 0 to 1. Figure 1 displays the technique utilized for pre-processing as well as the parameters of each of these steps.

### 3.3. Long Short-Term Memory

The choice of forecasting methodology used for time series depends on the features of the data and the specific forecasting goal [20]. ECG signals exhibit nonlinearity, nonstationarity, and temporal dependency features [5]. Statistical approaches often have limitations when dealing with data that have similar ECG characteristics and may not be able to capture all elements of time series [40,41]. LSTM models have been widely employed across various domains for time series forecasting, yielding favorable outcomes [25,26,32,44]. Some of the advantages of employing LSTM instead of a statistical technique include the following:LSTM models are very adept at capturing extended relationships in sequential data, which makes them highly useful for representing the complex patterns found in ECG signals;LSTM models are the best choice for handling intricate, nonlinear, nonstationary, and dynamic patterns, which are frequently observed in ECG signals;LSTM models have the ability to effectively manage sequences of varying lengths, allowing them to be easily applied to ECG recordings of diverse durations without the need for any pre-processing modifications.

However, while conducting our research, we discovered a constraint in ECG forecasting studies that have utilized this method. This discovery prompted us to investigate its suitability for these series.

Hochreiter and Schmidhuber [51] introduced LSTM in 1997 as a recurrent neural network with an exceptional capacity for learning and predicting sequential data. However, long-term training required the development and modification of LSTM to address vanishing and exploding gradient issues. Therefore, LSTM was designed to solve this constraint by adding a memory structure that can keep its state across time and gates to determine what to remember, forget, and output. In summary, LSTM outperforms alternative RNN designs by eliminating the vanishing gradient issue.

The LSTM architecture’s primary principles include a memory cell that can preserve its state over time and nonlinear gating components that control information flows in and out of the cell [52].

There are a few other LSTM architectures, each offering major or minor variations on the basic structure [53]. For this study, we employed vanilla LSTM, which was described in [52] as the LSTM variant that has been most frequently utilized in published works and is considered to be a de facto standard. Figure 2 illustrates a block diagram of the vanilla LSTM structure, which consists of multiple elements, including three gates (input, forget, and output), an input block, an output activation function, a peephole connection, and a memory block, named a cell.

The performance of LSTM may be greatly affected by the forget gate and the output activation function as performance suffers if one of them is removed. In order to keep a network’s learning from becoming unstable due to the unconstrained cell state, an output activation function is required [52].

#### 3.3.1. Forecasting Model

A total of 32,400 samples were chosen for each simulation, representing a duration of 90 s. Approximately 66.6% of these data were allocated for training purposes, while the remaining portion was designated for testing. It is important to note that, based on current understanding, there is no definitive definition for the optimal number of samples for certain subsets. Nevertheless, the prevailing method in the literature involves using the largest proportion of available data to train the model.

Time series forecasting is a predictive modeling technique that utilizes historical data to make projections about future occurrences. In order to create future predictions, it is important to establish the durations of the sequences of preceding occurrences. For the purposes of this investigation, the selected value was 720 samples. In addition to establishing the definitions of the sequences, it is important to establish the movements of the sequences across the series. To accomplish this in this study, a sliding window approach was used. Another parameter that must be specified is the quantity of forthcoming iterations to be forecasted. In this context, we adhered to the prevailing methodology that has been used in the literature, known as the one-step approach. Finally, the filtered signals were sent into the LSTM network. Figure 3 depicts the detailed structure of the LSTM model.

#### 3.3.2. Measuring Time Series Forecasting Performance

The mean absolute error (MAE) and root mean square error (RMSE) were utilized as assessment criteria to evaluate the prediction performance of the proposed model as they are the metrics that are generally recommended and used for forecasting in the literature [54].

The formula for calculating the MAE is as follows:(1)$$\mathrm{MAE}=\frac{\sum_{i=1}^{n}\left|y_{i}-x_{i}\right|}{n}$$
where $y_{i}$ represents the predicted value, $x_{i}$ represents the real value, and *n* denotes the total number of test set values.

The formula for calculating the RMSE is as follows:(2)$$\mathrm{RMSE}=\sqrt{\frac{1}{n}\sum_{i=1}^{n}{\left(Y_{i}-Y_{i}^{\prime }\right)}^{2}}$$

The closer MAE and RMSE values are to zero, the smaller the difference between the projected and actual values; therefore, the greater the accuracy of the forecasting.

## 4. Implementation

To implement the proposed model, we used a computer with the following characteristics:Operating system: Windows 11 Pro;Processor: 12th Gen Intel^®^ Core ™ i7-12,700 2.10 GHz;RAM: 64 GB (16*4 of DDR4);GPU: NVidia GeForce 1060—6 GB.

The code was written in Python 3.10.5 and developed with the help of the integrated development environment (IDE) PyCharm Community Edition 2022.1. A set of tools was utilized, specifically *wfdb 4.0.0a4* for accessing the database’s files via downloading and reading.

The iirnotch and lfilter functions from the scipy signal and the MinMaxScalar function from sklearn were all utilized in the pre-processing step. The values of the chosen parameters are presented in Figure 1 and the outcome is shown in Figure 4.

We utilized a *TensorFlow 2.9’s Keras* module to build the forecasting model. Imports included the sequential model, LSTM, Dense, and EarlyStopping Callbacks. The key features of the model are summarized in Figure 5 and its parameters are presented in Table 3. We evaluated the efficacy of the model by utilizing the inherent functionalities of the *math* module. *Matplotlib* was utilized to visually represent the pre-processing steps, loss functions, and graphs of the horizontal signals, as well as the training and test outcomes.

## 5. Results and Discussion

An ECG is a nonstationary, nonlinear, and nondeterministic time series, characteristics that are crucial for the effective development of forecasting models. Deep learning techniques have become viable alternatives for forecasting signals with such characteristics. Deep learning models demand a lot of computational power, and because the number of epochs of a term directly influences this, it is recommended to use a number of epochs that generates a good cost–benefit ratio. An example of a good strategy is the use of an early stopping criterion.

Despite implementing the initial proposal of 500 epochs, the model was stopped short of its full run-time after the loss function remained the same after 20 successive iterations, with an average value of 128.5±35.7 epochs. Figure 6 displays the number of training epochs. It is interesting to notice that most training was completed using fewer than 200 epochs. Despite this, the model’s performance was quite good, with just marginal amounts of inaccuracy.

For the presentation of the results, one of the database signals was chosen at random and the referent loss function of this signal is shown in Figure 7, where it can be seen that training was conducted over 119 epochs.

As described in Section 4, the RMSE and MAE metrics were used to evaluate the efficacy of the developed model. For the study in question, the RMSE was 0.0039 and the MAE was 0.0426. The RMSE and MAE values for each record in the dataset are displayed in Figure 8 and Figure 9, respectively. Table 4 presents the results by beat class, following Table 2. In addition, it is important to note that the aggregate results for the entire investigation were 0.0070±0.0028 for RMSE and 0.0522±0.0098 for MAE.

A case-specific outcome is depicted in Figure 10, which shows the behavior of an original signal along with the forecasted training and test signals. Figure 11 depicts a portion of the same signal to enhance its visibility.

It can be observed that the proposed model generated a forecasted signal with a similar pattern to that of the original signal, thereby enabling the prediction of fundamental variations in the signal.

Due to the fact that an entire ECG can be observed in approximately 1 s, the prognostic procedures utilized had a direct effect on the final findings. This effect was typically positive since the differences between the predicted values and the actual series were so minor.

Table 4 demonstrates that the accuracy of the predictions across assessments. However, it also indicates that the results were comparable. This was largely due to the fact that (a) the levels of noise and interference in the initial signals were not the same, so in some cases, interference could persist after pre-processing, which could have influenced the model, and (b) the data exhibited temporal inconsistencies as a result of being captured with various anomalies. Nevertheless, we can assert that the proposed model is effective for various signals with comparable properties.

Table 5 illustrates a comparative analysis of the outcomes from prior research papers that used identical measures and data from the same database.

Based on the data shown in the table, it is noticeable that although our research was conducted using the same dataset as the others, none of the previous studies included all simulations in their experimental designs. In contrast, the technique under consideration demonstrated superior outcomes compared to those from prior research endeavors, particularly the study conducted by Dudikcu [45], which similarly explored deep learning-based approaches. The findings of this study provide more support for the notion that deep learning models provide favorable predictive performance owing to their capacity to process both short-term and long-term signals effectively.

The relevance of ECG signals within the healthcare industry is well known. Therefore, the proposed model could be used in the following situations:The early detection of cardiac abnormalities as ECG signal forecasting can detect heart irregularities and potentially fatal illnesses; therefore, ECG data can be used to diagnose cardiac anomalies, such as arrhythmia, ischemia, heart obstructions, and more, and early diagnosis can lead to better patient outcomes;ECG signal forecasting aids in patient risk assessment and stratification as by studying ECG data and identifying problematic patterns or indications, healthcare providers can categorize patients and this information can help with resource allocation and treatment decisions;Doctors can better comprehend a patient’s cardiac condition by predicting ECG patterns as by using a patient’s historical ECG data, forecasting algorithms can predict future trends and identify changes in heart function, which enables tailored therapy and interventions;ECG signal forecasting is required for remote monitoring and telemedicine, so healthcare providers can remotely monitor patients with persistent heart conditions by continuously monitoring ECG data, which simplifies cardiac health management while also lowering healthcare costs and increasing patient convenience;ECG signal forecasting aids in determining the success of cardiac treatment, so healthcare providers can assess a patient’s cardiac health by comparing expected ECG patterns to actual data, which can improve treatment strategies and guide ongoing care;ECG signal forecasting provides information on long-term cardiac health, so ECG data can help healthcare personnel to monitor cardiac issues, treatments, and treatment plans as the R-peaks have larger amplitudes than the other waves in the signals, which are where the most noticeable changes may be seen.

## 6. Conclusions

According to the results presented, the following conclusions about the LSTM ECG signal forecasting model could be drawn:The forecasting of ECG signals is, to a large extent, a rather unexplored field and this is one of the first papers to employ deep learning techniques (LSTM) to the forecasting of ECG time series;The proposed LSTM model can predict the trends of changes in original data series (with the most visible differences being in the amplitudes of the R-peaks), which shows that the LSTM model can meet the requirement of forecasting accuracy;Considering the accuracy of the proposed model and taking into account the properties of the physiological signals, a similar approach could be applied to ECG signals and help to improve the efficiency of healthcare systems;Research within the field of ECG forecasting remains minimal, which allows for the development of other studies, such as the analysis of forecasting steps, the study of different forecasting strategies, or even the investigation of applications based on other methods.

### Future Work

This study contributes to the field of ECG signal forecasting; nevertheless, several areas still need more investigation and potential improvements. In this subsection, we provide a comprehensive overview of potential directions for future studies.

The research reported in this paper and most of the existing literature adopted a one-step forecasting technique. However, the domain of multistep forecasting offers many opportunities for future research and exploration.

One identified challenge pertains to effectively considering the spatial and temporal aspects of signals during the forecasting procedure. Consequently, future efforts in spatiotemporal forecasting are necessary to mitigate the observed disparities in signal amplitudes. These findings were noticed in our investigation.

The development of real-time systems that are capable of using predictive analysis to anticipate cardiac abnormalities, with a particular focus on cardiac arrhythmia, is equally important.

## Figures and Tables

**Figure 1 bioengineering-11-00089-f001:**
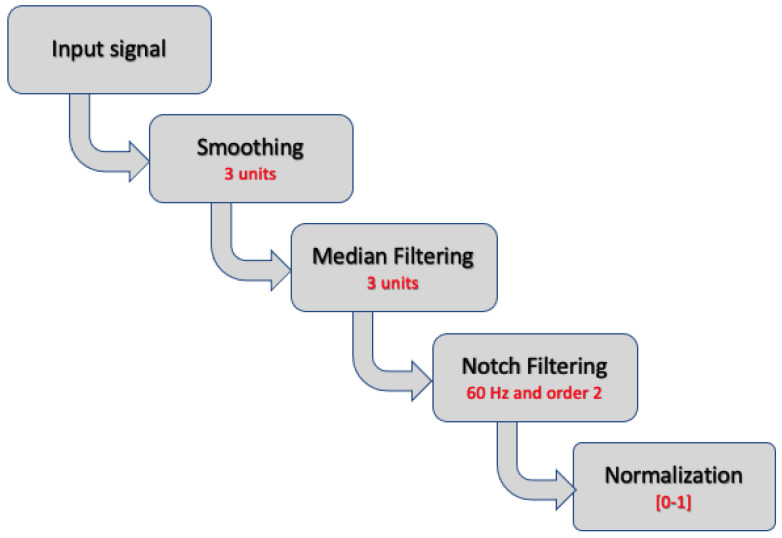
Procedures followed in the pre-processing stage.

**Figure 2 bioengineering-11-00089-f002:**
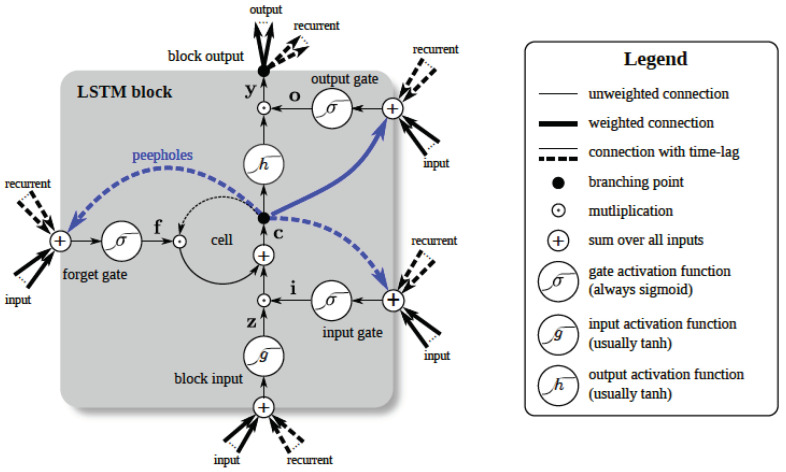
Schematic diagram of the *vanilla* LSTM structure (Adopted with permission from [52], Copyright 2023 IEEE).

**Figure 3 bioengineering-11-00089-f003:**
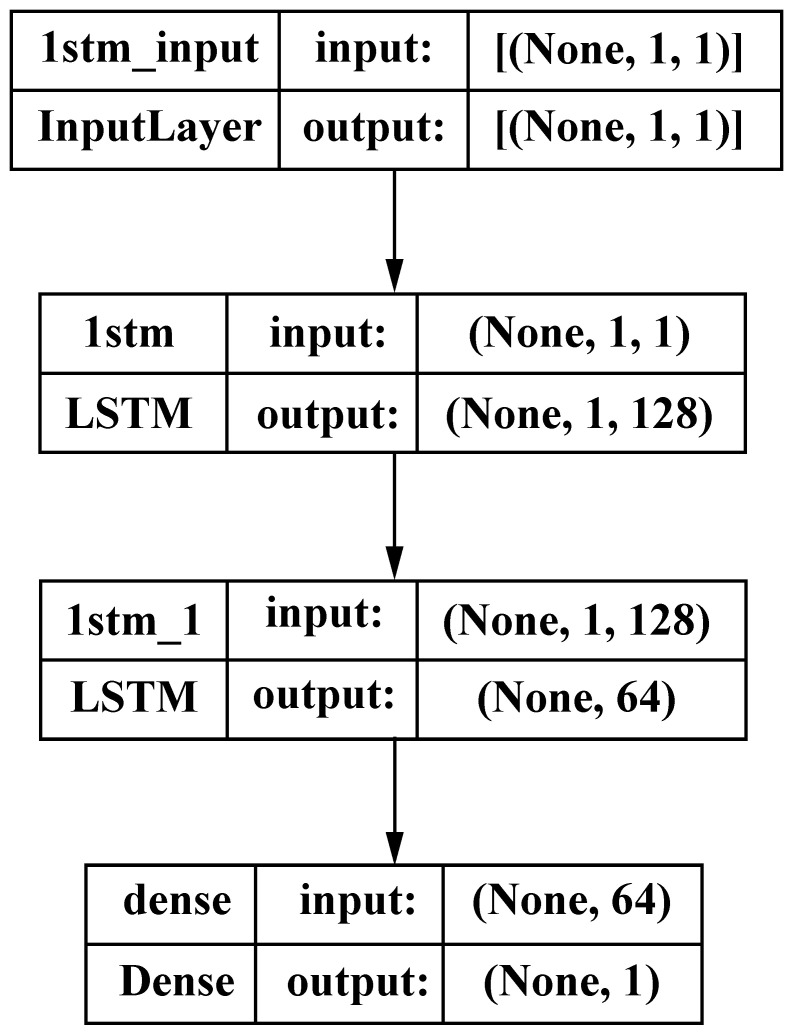
Proposed LSTM network architecture.

**Figure 4 bioengineering-11-00089-f004:**
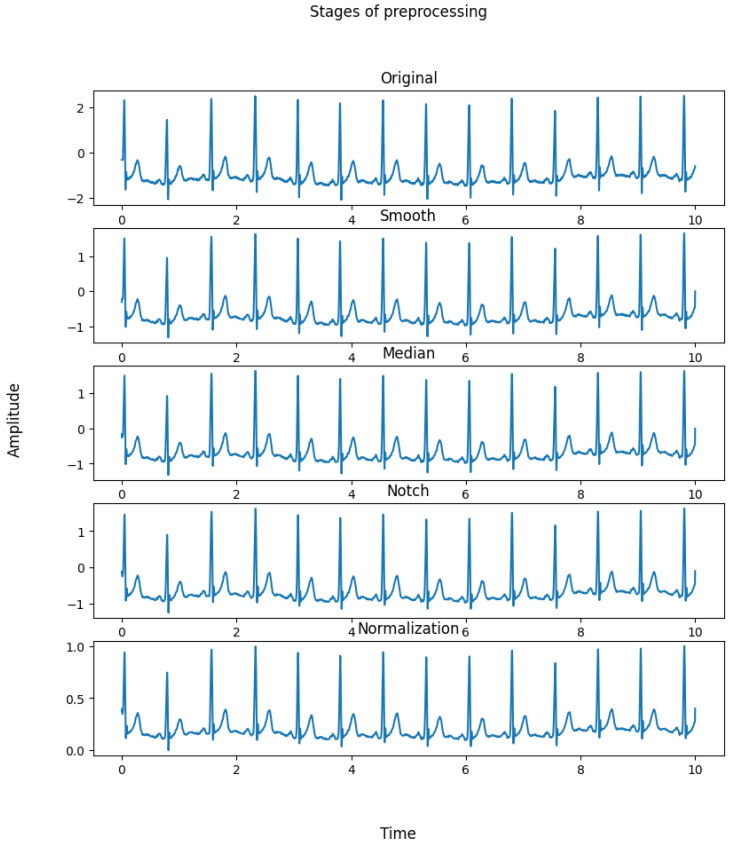
Pre-processing steps for electrocardiogram signals.

**Figure 5 bioengineering-11-00089-f005:**
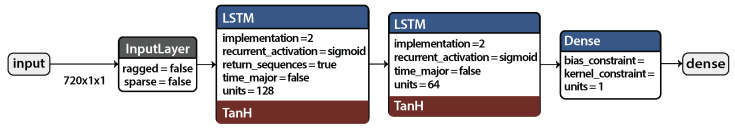
Summary of the implemented network.

**Figure 6 bioengineering-11-00089-f006:**
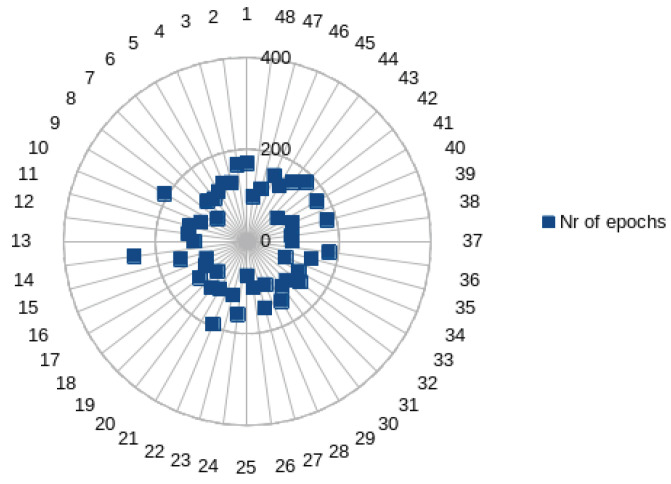
Number of training epochs for each signal.

**Figure 7 bioengineering-11-00089-f007:**
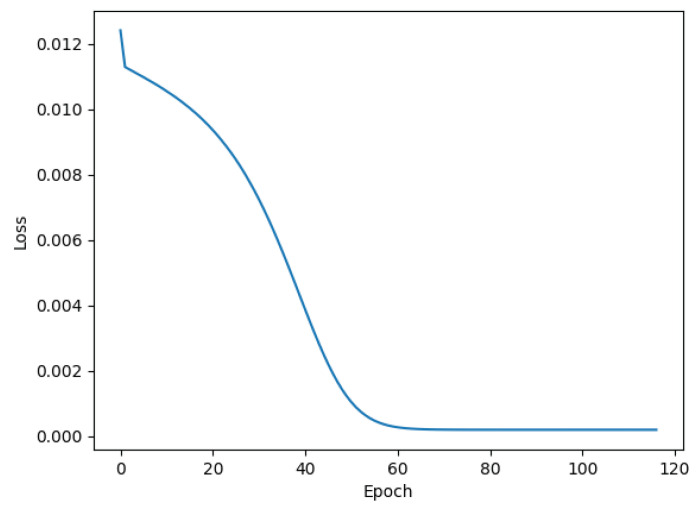
Graph of the loss function over the epochs.

**Figure 8 bioengineering-11-00089-f008:**
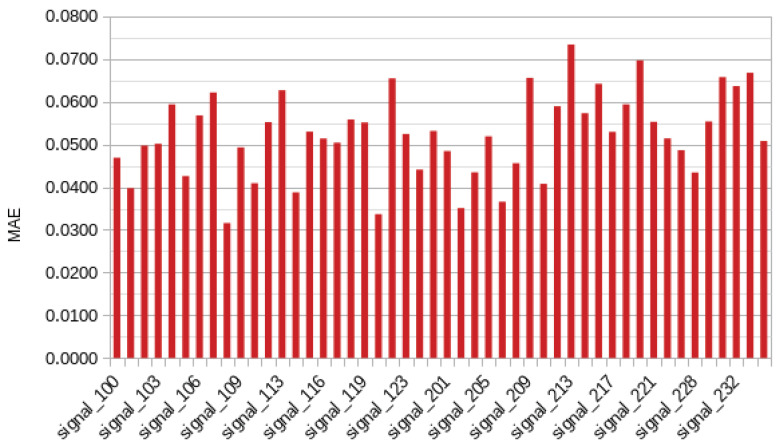
Mean absolute error of each record.

**Figure 9 bioengineering-11-00089-f009:**
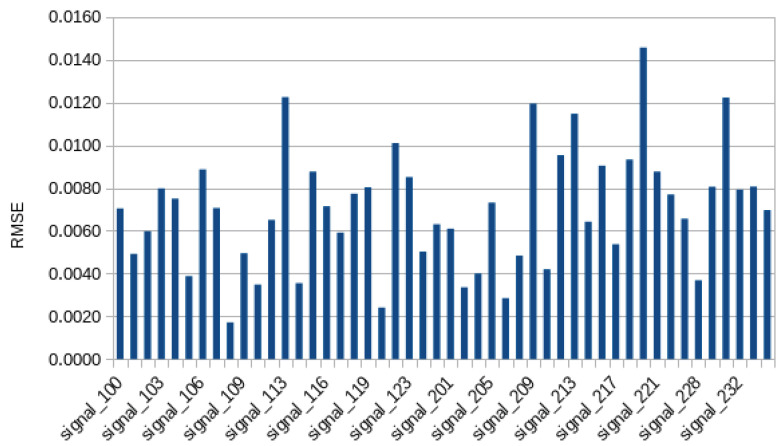
Root mean square error of each record.

**Figure 10 bioengineering-11-00089-f010:**
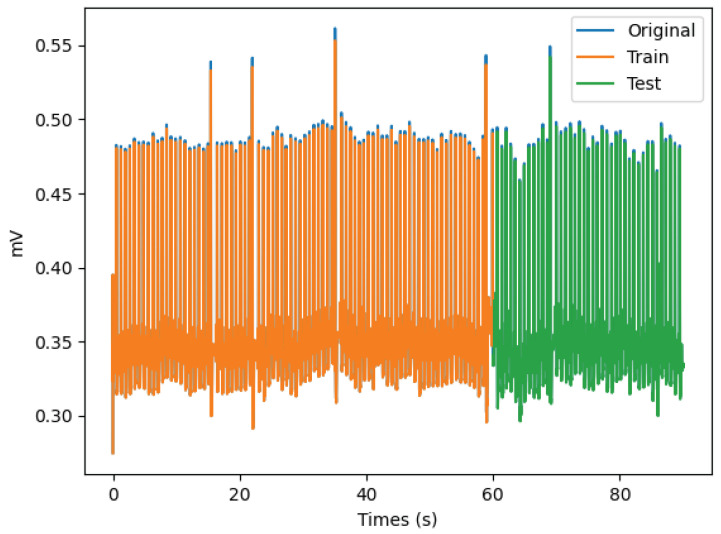
Presentation of an example signal, with the original, training and test signals.

**Figure 11 bioengineering-11-00089-f011:**
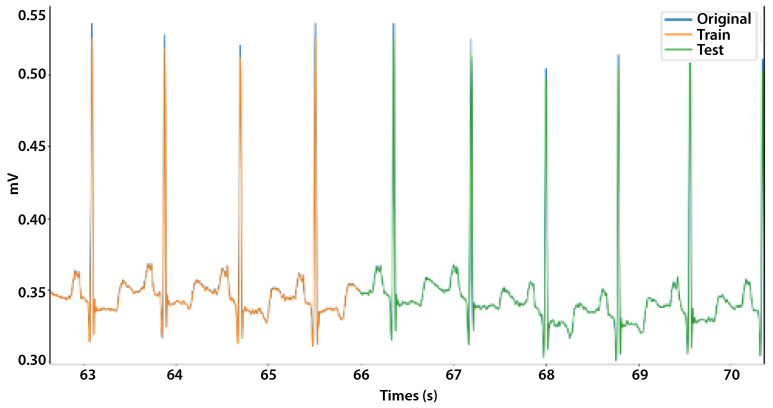
Sample of a forecasted ECG signal. The blue represents the original signal, the orange represents the training signal, and the green represents the test signal.

**Table 1 bioengineering-11-00089-t001:** Summary of the studies found in the presented literature review.

Study	Forecasting Technique	Database	Sequence Length	Forecasting Steps	Metric			
					MAE	MSE	RMSE	$R^{2}$
Sun et al. [40]	VMD and BPNN	MIT-BIH	N.S	1	0.016	x	0.023	x
Sun et al. [41]	MI and BPNN	MIT-BIH	N.S	1	0.02	x	0.042	x
Huang et al. [42]	ARIMA and DWT	MIT-BIH	N.S	1	0.011	x	0.018	x
Mohammadi et al. [39]	EWT and HFCM	N.F	N.S	1	x	x	0.011	x
Festag et al. [46]	rcGAN	Autonomic Aging Physio net	1250	1	x	0.043	x	x
				30	x	0.086	x	x
				250	x	0.090	x	x
Prakarsha et al. [43]	ANN	Simulated Data and Pyshio net ATM of Sleep Apnea	64	1	0.045	x	x	x
	LMS				0.21	x	x	x
Dudukcu et al. [45]	TCN-LSTM	MIT-BIH	10	1	0.005	x	0.008	0.991
	TCN-GRU				0.005	x	0.008	0.990

N.S, not specified; N.F, not found.

**Table 2 bioengineering-11-00089-t002:** Distribution of ECG records in the MIT-BIH database.

Type	Records
Normal beat	100, 101, 103, 105, 108, 112, 113, 114, 115, 117, 121, 122, 123, 202, 205, 219, 230, 234, 109, 111, 207, 214.
LBBB beat	109, 111, 207, 214.
RBBB beat	118, 124, 212, 231.
PVC beat	106, 116, 119, 200, 201, 203, 208, 213, 221, 228, 233.
APB beat	209, 220, 222, 223, 232.

**Table 3 bioengineering-11-00089-t003:** Parameters and hyperparameters of the proposed method.

Parameter	Value
Batch size	16
LTSM unit	192
Optimizer	SGD
Max number of epochs	500
Training/test	66/34
Loss	Mean squared error
Activation	than
Metric	MAE
Early stopping	Patience = 20; mode = min; monitor = loss
Number of hidden LSTM layers	2
Average number of epochs	130

**Table 4 bioengineering-11-00089-t004:** Model performance by beat class.

Type	Metric	
	MAE	RMSE
Normal beat	0.007 ± 0.0027	0.0506 ± 0.0093
LBBB beat	0.0042 ± 0.0016	0.0451 ± 0.0092
RBBB beat	0.0086 ±0.0030	0.05735 ± 0.0090
PVC beat	0.0071 ± 0.0024	0.0531 ± 0.0094
APB beat	0.0081 ± 0.0034	0.0656 ± 0.0097

**Table 5 bioengineering-11-00089-t005:** Comparison of results from the literature and those from the proposed method.

Study	Forecasting Technique	Number of ECG Signals	Sequence Length	MAE	RMSE
Sun et al. [40]	VMD and BPNN	1	Non	0.0157	0.0233
Sun et al. [41]	MI and BPNN	1	Non	0.024	0.0423
Huang et al. [42]	ARIMA and DWT	4	Non	0.0111	0.0181
Dudukcu et al. [45]	TCN-LSTM	21	10	0.0051	0.0082
	TCN-GRU			0.0052	0.0084
Our study	LSTM	47	720	0.0522 ± 0.01	0.0070 ± 0.003

## Data Availability

In this study, an open-access dataset (MIT-BIH) sourced from Physio net was used, which may be accessed via the provided URL https://physionet.org/content/mitdb/1.0.0/, accessed on 25 July 2023.

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
