# Peer review of "ECG Forecasting System Based on Long Short-Term Memory"

_bioengineering, 2024, doi:10.3390/bioengineering11010089_

Round 1

Reviewer 1 Report

Comments and Suggestions for Authors

Manuscript ID: bioengineering-2737079 Manuscript Type: Full Length Article

Title: ECG forecasting system based on Long Short-Term Memor

Comments to the Authors

I reviewed the manuscript, “bioengineering-2737079; ECG forecasting system based on Long Short-Term Memor.” with great pleasure.

The objective of this study is to develop a model using deep learning techniques for the long-term prediction of electrocardiogram (ECG) signals, thereby aiding in the early detection and diagnosis of cardiovascular diseases. Specifically, this model is based on Long Short-Term Memory (LSTM) architecture, learning the non-linearity, non-stationarity, and complexity of ECG signals to predict future ECG patterns. The aim is to enable medical professionals to predict the condition of patients based on diagnoses at very early stages and to administer appropriate treatment accordingly. Furthermore, this model has the potential to enhance existing predictive healthcare systems focused on cardiovascular monitoring.

An ECG represents the electrical activity of the heart, and the patterns it exhibits can vary significantly depending on the type of disease. The use of deep learning for predicting ECG waveforms presents a fascinating area of study; however, there are several issues that warrant discussion.

1.      Certain diseases, such as arrhythmias and myocardial infarction, are known to demonstrate specific changes in the ECG. In contrast, other diseases may show less pronounced or non-specific changes on the ECG. Therefore, in the development of predictive models using ECG data, it is crucial to clearly define the type of disease or cardiac condition being targeted. If a model is specialized in predicting the ECG patterns of a specific disease, it may be highly useful for that particular condition but not necessarily appropriate for others. Non-specific alterations are often observed in ECG in conditions other than certain cardiac diseases. Consequently, the efficacy of the model is contingent on the disease-specific characteristics of the utilized ECG waveforms. To substantiate the utility of the authors' proposed model for ECG waveform prediction, it is imperative to employ disease-specific raw data pertaining to specific cardiac conditions.

2.      ECG findings can be influenced by various physical factors, including age, gender, and the time of examination, with these changes often being non-specific. This consideration is crucial when analyzing ECG data and developing predictive models. Ideally, in the analysis of ECG data, especially when employing machine learning or deep learning techniques, it is important that the model is trained to appropriately distinguish the effects of these varying factors.

3.      The current manuscript is extensive, lacks coherence, and poses challenges in comprehension. It is advisable to restructure it in a systematic format, clearly delineating the study's objectives, targets, methodology, findings, and discussion for enhanced clarity and readability."

Comments on the Quality of English Language

Average

Author Response

  1. Certain diseases, such as arrhythmias and myocardial infarction, are known to demonstrate specific changes in the ECG. In contrast, other diseases may show less pronounced or non-specific changes on the ECG. Therefore, in the development of predictive models using ECG data, it is crucial to clearly define the type of disease or cardiac condition being targeted. If a model is specialized in predicting the ECG patterns of a specific disease, it may be highly useful for that particular condition but not necessarily appropriate for others. Non-specific alterations are often observed in ECG in conditions other than certain cardiac diseases. Consequently, the efficacy of the model is contingent on the disease-specific characteristics of the utilized ECG waveforms. To substantiate the utility of the authors' proposed model for ECG waveform prediction, it is imperative to employ disease-specific raw data pertaining to specific cardiac conditions.

    R: The reviewer comment is useful, but, it focuses on diagnostic. Our primary focus is to assess the changes that are subtle in the ECG waveform, and most particularly, in the stage of pre-diagnostic. Diagnostic is important for therapeutic and prognostic. We can deal with diagnostic after the disease has been identified, we want to catch the changes in the signal before the diagnostic.

  2. ECG findings can be influenced by various physical factors, including age, gender, and the time of examination, with these changes often being non-specific. This consideration is crucial when analyzing ECG data and developing predictive models. Ideally, in the analysis of ECG data, especially when employing machine learning or deep learning techniques, it is important that the model is trained to appropriately distinguish the effects of these varying factors.

    R: Working with existing databases allows us to not consider the underlying demographics. Once the concept is demonstrated for a particular set of signals, we can work on extending it to other age intervals, and so on.

  3. The current manuscript is extensive, lacks coherence, and poses challenges in comprehension. It is advisable to restructure it in a systematic format, clearly delineating the study's objectives, targets, methodology, findings, and discussion for enhanced clarity and readability."

    R: We thank the reviewer for this comment. We have revised the manuscript in order to improve its readability.

Reviewer 2 Report

Comments and Suggestions for Authors

1. In line 55, is it possible to confirm the presence of a question mark (?) in the reference?Some references also contain a question mark (?). Please have the author confirm this thoroughly.

2. What method is used for smoothing in the preprocessing?

3.In the graph of the loss function over epochs in the text, may I inquire about the reason for setting the number of epochs to 120?

Author Response

  1. What method is used for smoothing in the preprocessing?

    R. The authors failed to mention the specific method employed, but expressed gratitude for the correction and subsequently revised the text to indicate the utilization of the simple moving average (SMA) smoothing approach.

  2. In the graph of the loss function over epochs in the text, may I inquire about the reason for setting the number of epochs to 120?

    R: The configuration specified 500 epochs and a halting criterion was created as outlined in lines 217 and 218. The mean number of epochs is 128.5 ± 35.7, as indicated in line 219. The figure depicts a specific instance where 120 epochs were required based on the halting condition.

  3. In line 55, is it possible to confirm the presence of a question mark (?) in the reference?Some references also contain a question mark (?). Please have the author confirm this thoroughly.

    R. The presence of question marks within the text can be attributed to an unsuccessful update of the bibliography file. Nevertheless, we express thankfulness for the repair conducted, and we have duly revised and updated the text.

Reviewer 3 Report

Comments and Suggestions for Authors

Interesting paper. Here are few suggestions for improvements:

What is the "?" doing after reference 4 in line 48? Same issue in line 55 (i.e., ? appeared after reference 9). Are you not sure about these references? If not, then remove them and replace with more updated references.

Section 2 (literature review) is not comprehensive and clear. The author should first highlight that ECG analysis using machine learning technique is an age old problem and researchers had been using MIT-BIH database to evaluate their machine learning algorithms starting from more than a decade ago. In this regards, one or more of previous studies like following could be cited:

1.       Fahim Sufi, Ibrahim Khalil, Abdun Mahmood, “A clustering based system for instant detection of cardiac abnormalities from compressed ECG”, Expert Systems with Applications (Elsevier), Volume 38, Issue 5, pp 4705-4713, 2011 2011

2.       Fahim Sufi and Ibrahim Khalil, Diagnosis of cardiovascular abnormalities from Compressed ECG: A Data Mining based Approach, IEEE Transaction in Information Technology in Biomedicine, Volume 15, Issue 1, pp. 33-39, 2011

Moreover, literature review should be written in a way that clearly articulate the deficiency of existing research. To do that I suggest creating presenting the existing studies in ECG prediction and their corresponding accuracies / MSE / RMSE / MAE in a TABULAR manner (The first part (i.e., without the result of this study) of Table 4 could be moved here). This table would allow the readers to quickly grasp the existing disadvantages and possible improvements (i.e., justification of this study). You can clearly highlight that accuracy, number of evaluating ECG, Sequence length as possible areas of improvements (i.e., the focus of this innovative method).

Literature review or methodology section should suggest the justification and rational behind selecting LSTM as opposed to other forecasting techniques (e.g., exponential smoothing etc.).

Within section 4 (Implementation), the authors need to portray the name of tools and their purpose in a tabular manner. Only mentioning "multiple tools were used" (i.e., in line 237) goes against research reproducibility principal.

Author Response

  1. What is the "?" doing after reference 4 in line 48? Same issue in line 55 (i.e., ? appeared after reference 9). Are you not sure about these references? If not, then remove them and replace with more updated references.

    R. The presence of question marks within the text can be attributed to an unsuccessful update of the bibliography file. Nevertheless, we express thankfulness for the repair conducted, and we have duly revised and updated the text.

  2. Section 2 (literature review) is not comprehensive and clear. The author should first highlight that ECG analysis using machine learning technique is an age old problem and researchers had been using MIT-BIH database to evaluate their machine learning algorithms starting from more than a decade ago. In this regards, one or more of previous studies like following could be cited:

    1. Fahim Sufi, Ibrahim Khalil, Abdun Mahmood, “A clustering based system for instant detection of cardiac abnormalities from compressed ECG”, Expert Systems with Applications (Elsevier), Volume 38, Issue 5, pp 4705-4713, 2011 2011

    2. Fahim Sufi and Ibrahim Khalil, Diagnosis of cardiovascular abnormalities from Compressed ECG: A Data Mining based Approach, IEEE Transaction in Information Technology in Biomedicine, Volume 15, Issue 1, pp. 33-39, 2011

      R: We updated the section 2 (literature review) in order to be comprehensive and Clear

  3. Moreover, literature review should be written in a way that clearly articulate the deficiency of existing research. To do that I suggest creating presenting the existing studies in ECG prediction and their corresponding accuracies / MSE / RMSE / MAE in a TABULAR manner (The first part (i.e., without the result of this study) of Table 4 could be moved here). This table would allow the readers to quickly grasp the existing disadvantages and possible improvements (i.e., justification of this study). You can clearly highlight that accuracy, number of evaluating ECG, Sequence length as possible areas of improvements (i.e., the focus of this innovative method).

    R: The update to the literature review section also considers this comment.

  4. Literature review or methodology section should suggest the justification and rational behind selecting LSTM as opposed to other forecasting techniques (e.g., exponential smoothing etc.).

    R: We thank the reviewer for this comment. We update the methodology section to present the justification and rational behind selecting LSTM as opposed to other forecasting techniques.

  5. Within section 4 (Implementation), the authors need to portray the name of tools and their purpose in a tabular manner. Only mentioning "multiple tools were used" (i.e., in line 237) goes against research reproducibility principal.

    R. We thank the reviewer for this comment. We revised the manuscript to ensure greater reproducibility.

Round 2

Reviewer 3 Report

Comments and Suggestions for Authors

The authors have addressed all my previous concerns. Only issue is the author should properly format Table 1 so that it appears within the page boundary. Otherwise, I am happy with the updated manuscript. 

Author Response

Esteemed reviewer, We appreciate your notification about the table format. The document was revised to adhere to the correct formatting.